# Ionospheric Responses to the June 2015 Geomagnetic Storm from Ground and LEO GNSS Observations

**Chao Gao [1,2], Shuanggen Jin [1,3,4],\* and Liangliang Yuan [1,2]**

1. Shanghai Astronomical Observatory, Chinese Academy of Sciences, Shanghai 200030, China; gaochao17@mails.ucas.edu.cn (C.G.); llyuan@shao.ac.cn (L.Y.)
2. School of Astronomy and Space Science, University of Chinese Academy of Sciences, Beijing 100049, China
3. School of Remote Sensing and Surveying Engineering, Nanjing University of Information Science and Technology, Nanjing 210044, China
4. Jiangsu Engineering Center for Collaborative Navigation/Positioning and Smart Applications, Nanjing 210044, China
\* Correspondence: sgjin@shao.ac.cn; Tel.: +86-21-34775292

**Abstract:** Geomagnetic storms are extreme space weather events, which have considerable impacts on the ionosphere and power transmission systems. In this paper, the ionospheric responses to the geomagnetic storm on 22 June 2015, are analyzed from ground-based and satellite-based Global Navigation Satellite System (GNSS) observations as well as observational data of digital ionosondes, and the main physical mechanisms of the ionospheric disturbances observed during the geomagnetic storm are discussed. Salient positive and negative storms are observed from vertical total electron content (VTEC) based on ground-based GNSS observations at different stages of the storm. Combining topside observations of Low-Earth-Orbit (LEO) satellites (GRACE and MetOp satellites) with different orbital altitudes and corresponding ground-based observations, the ionospheric responses above and below the orbits are studied during the storm. To obtain VTEC from the slant TEC between Global Positioning System (GPS) and LEO satellites, we employ a multi-layer mapping function, which can effectively reduce the overall error caused by the single-layer geometric assumption where the horizontal gradient of the ionosphere is not considered. The results show that the topside observations of the GRACE satellite with a lower orbit can intuitively detect the impact caused by the fluctuation of the F2 peak height (hmF2). At the same time, the latitude range corresponding to the peak value of the up-looking VTEC on the event day becomes wider, which is the precursor of the Equatorial Ionization Anomaly (EIA). However, no obvious response is observed in the up-looking VTEC from MetOp satellites with higher orbits, which indicates that the VTEC responses to the geomagnetic storm mainly take place below the orbit of MetOp satellites.

**Keywords:** geomagnetic storm; Ionosphere; VTEC; GRACE; MetOp

## 1. Introduction

The geomagnetic storm is a severe global disturbance of the Earth's magnetic field, usually under the impact of disturbances in the solar wind and Interplanetary Magnetic Field (IMF) with their origins near the solar surface. Extensive studies have shown that intense geomagnetic storms are usually caused by coronal mass ejections (CMEs) [1]. Geomagnetic storms result in an increase of the ring current, and the magnetic field generated by the ring current is superimposed on the geomagnetic field, which causes great changes in the horizontal component H. When a geomagnetic storm occurs, Earth's entire magnetic field will be continuously disturbed for several hours to tens of hours, and all geomagnetic elements will undergo drastic changes. At the same time, geomagnetic storms are often accompanied by the transfer and conversion of energy in the magnetosphere, such as the generation

of various waves and particle acceleration in the plasma, etc. These changes will have direct and indirect impacts on satellites operating in the magnetosphere and ionosphere (e.g., Geostationary, Medium and Low-Earth orbits), such as radiation hazard to electronics and communications through the ionosphere. The high-energy electrons will increase significantly after intense geomagnetic storms, causing deep charging of satellites. In addition, the geomagnetic-induced currents (GIC) generated by the disturbance of the Earth's magnetic field will also affect the high-voltage power equipment on the ground.

The phenomenon of dramatic changes in the ionosphere during geomagnetic storms is called an ionospheric storm. Ionospheric storms can be divided into positive and negative storms according to the change of the peak electron density relative to the reference value in the quiet period [2]. Ionospheric storms have been a hot topic in the field of Earth and space science since they were first detected in the mid-20th century [3,4]. As for the physical mechanism of ionospheric storms, different scholars have put forward different theories, among which the theory of thermospheric storm circulation proposed by Duncan and continuously improved by many scholars is considered as the main formation mechanism of ionospheric storms (especially for positive storms), which can well explain many characteristics of mid-latitude ionospheric storms [5–8].

Constrained by the structure of the Earth's magnetic field, the equatorial and low-latitude ionosphere is mainly affected by two types of electric fields during geomagnetic storms. The eastward prompt penetration electric field (PPEF) associated with magnetospheric convection with a relatively short duration of one to two hours [9], and the westward disturbance dynamo electric field (DDEF) with a duration of generally more than three hours [10]. Many important results have been obtained through studies on super geomagnetic storm events. During intense geomagnetic storms, the equatorial ionosphere will be uplifted by PPEF, resulting in the "super fountain effect" [11]. The ionospheric F3 layer on the vertical structure is formed, and the responses of the top and bottom ionosphere are significantly different [12,13]. In addition, the PPEF occurring at dusk can enhance the Rayleigh Taylor instability of the ionospheric plasma and trigger obvious ionospheric F-layer irregularities and ionospheric scintillations in the equatorial, low-latitude, and even mid-latitude regions [14]. In the daytime, the westward DDEF will suppress the fountain effect in the equatorial region, resulting in positive storms in the equatorial region and negative storms in the low latitude region, which usually occurs during the recovery phase of geomagnetic storms and lasts for one to two days. In addition to the influence of electric field, the drift caused by the meridional wind during a geomagnetic storm will also prevent the plasma from diffusing along the magnetic field lines to the low-latitude region and will inhibit the formation of the equatorial bimodal, thus resulting in a negative storm in the low-latitude region and a positive storm in the equatorial trough region [15].

With regard to ionospheric responses to geomagnetic storms, ionospheric parameters such as the total electron content (TEC), the F2 peak electron density (NmF2), and the F2 peak height (hmF2) were normally used [16]. A digital ionosonde can continuously monitor the ionosphere above the station with the hmF2 and NmF2 parameters of the ionospheric F2 layer, which are important observational data for studies on ionospheric magnetic storms [17]. Based on the dispersion characteristics of the ionosphere, TEC on the signal propagation path can be obtained by using dual-frequency Global Navigation Satellite Systems (GNSS) observations [18]. There are a lot of advantages in using ionospheric TEC by GNSS observations, such as global coverage, high spatial-temporal resolution, high measurement accuracy, and so on. At present, GNSS observations have been widely used in studies on ionospheric geomagnetic storms [19,20]. However, most studies used ground- or satellite-based GNSS observations during a geomagnetic storm in the past, while ionospheric responses at different altitudes are not clear.

Based on ground-based and satellite-based GNSS observations as well as observational data of digital ionosondes, the ionospheric response to the intense geomagnetic storm on 22 June 2015 is studied in this paper. The distribution of ionospheric vertical TEC (VTEC) in the low- and mid-latitude regions along ~120°E (110°E–130°E, 60°S–60°N) is obtained by ground-based GNSS observations during the intense geomagnetic storm. The variation of the VTEC difference between storm days and

quiet days is analyzed. In the study region, the ionospheric parameters hmF2 and NmF2 during the storm days are obtained from four uniformly distributed Ionosonde stations in China, which are Sanya, Wuhan, Beijing, and Mohe stations. The parameters are also compared with the reference values of quiet days. At the same time, the topside GNSS observations of Low-Earth-Orbit (LEO) satellites with different orbital altitudes are also used. In order to precisely convert the slant TEC of the GPS-LEO path into VTEC, we employed a multi-layer mapping function. Through combination of the topside observations of LEO satellites with the ground-based GNSS observations in corresponding time and space, variations of VTEC with different altitude ranges throughout the whole storm event are studied, and the possible causes and physical mechanisms of the variations are explained. In Section 2, data and methods are introduced, results and analysis are presented in Section 3, and finally conclusions and discussion are given in the last section.

## 2. Data and Methods

### 2.1. Observational Data

The data used in our study include ground-based observations and satellite-based observations. The ground-based observations include Ionosonde observational data and ground-based GNSS observations. The Ionosonde observational data are selected from four stations in China, which are relatively close in longitude and uniformly distributed in latitude. The stations are Sanya station (109.4°E, 18.3°N), Wuhan station (114.6°E, 30.5°N), Beijing Ming Tombs station (116.2°E, 40.3°N), and Heilongjiang Mohe station (122.4°E, 53.5°N). The time resolution of the observational data is 15 min, which can be obtained from the Center for Space Environment Research and Forecast (CSERF) of Chinese Academy of Sciences (http://www.cserf.ac.cn/download/index.php). The global ionospheric maps (GIMs) provided by the international GNSS service (IGS) are directly used as ground-based GNSS observations in this paper. The satellite-based observations are the topside observations of GRACE and MetOp LEO satellites, and the corresponding time period of the observations is from June 20 to 24, 2015. At the same time, the ground-based observations from June 15 to 20 are used to provide reference values for the geomagnetic calm days.

Since 1998, IGS has been providing global ionospheric products with a spatial resolution of 5° in longitude and 2.5° in latitude and a time resolution of two hours for global ionospheric research and applications [21,22]. At present, there are several institutions with providing ionospheric products to IGS, including the Centre for Orbit Determination in Europe (CODE), Jet Propulsion Labs (JPL), European Space Agency (ESA), Universitat Politècnica de Catalunya (UPC), Chinese Academy of Sciences (CAS), and so on. Here, we selected GIM products provided by UPC for this study, since the temporal resolution is higher (15 min) and the resolution is the same as that of Ionosonde measurements. Observational data of about 300 GPS ground-tracking stations around the world were used for modeling, with a sampling rate of 30 s and a satellite cut-off altitude angle of 10°. The phase-smoothed pseudorange was used to calculate TEC between GPS satellites and the ground receivers, and the global VTEC model was established by using 15-order spherical harmonic functions. The errors of GIM TEC products include mapping function error, differential code biases (DCBs), pseudorange, and noise errors, etc. [23]. According to the evaluation results of the IGS Ionospheric Associated Validation Center (IAVC), the average deviation of UPC GIM products is about 1 TECU, and the standard deviation is about 4 TECU [24], which meets the accuracy requirements for analyzing the variation of ionospheric VTEC during the intense geomagnetic storm.

The satellite-based observations used in this paper are the topside observations of GRACE and MetOp satellites, which can be obtained from the COSMIC Data Analysis and Archive Center (CDAAC) (https://www.cosmic.ucar.edu/cdaac/). The GRACE satellite was developed by National Aeronautics and Space Administration (NASA) in cooperation with the German Aerospace Center (DLR) and was launched in 2002. Its constellation consists of two LEO satellites operating at the same near-polar orbit. In June 2015, the orbital altitude was about 400 km and the inclination angle was 89°. In our research

period, however, observations of GRACE-B satellite could not be used and were eliminated in our data processing. The European polar orbiting meteorological satellite MetOp also consists of two LEO satellites, i.e., MetOp-A and MetOp-B, both of which also operate at the same orbit, with an orbital altitude of about 820 km and an inclination angle of 98.7°.

The Coordinated Universal Time (UTC), three-dimensional coordinates of LEO and GPS satellites, observational elevation of the GPS-LEO link at LEO satellite, and the slant TEC on the signal propagation path at each epoch of observation are recorded in a podTec file. The slant TEC is the absolute value of TEC in which the DCBs of the GPS satellite and receiver on LEO satellite are removed [25]. The time resolutions of the topside observations of GRACE and MetOp satellites are 10 s and 1 s, respectively, and we resampled the observations of MetOp satellites into a resolution of 10 s in data processing. The altitude ranges of the topside observations of GRACE and MetOp satellites are from the orbital altitude of LEO satellites to the orbital altitude of GPS satellites, i.e., 400–20,200 km and 820–20,200 km for GRACE and MetOp satellites, respectively. Since the TEC data are the slant TEC on the signal propagation path, VTEC time series in the zenith direction are obtained by employing a multi-layer mapping function [26], which is introduced in the next section.

## 2.2. Multi-Layer Mapping Function

When modeling the ionospheric TEC, such as the GIM, the ionospheric single-layer hypothesis is usually used, which assume that the ionospheric electrons are concentrated on a spherical shell of infinitesimal thickness at a fixed altitude surrounding the Earth, and the typical altitude of the spherical shell is 350–480 km [27]. Then, a mapping function of the observational elevation angle is used as the conversion factor to convert the slant TEC into VTEC. The most commonly used mapping functions include the cosine mapping function and F&K mapping function [28,29]. The assumption of a single layer is simple and practical, but its shortcomings are also obvious. Since the space–time characteristics of the ionosphere, including a horizontal gradient and vertical structure, are not considered, the single-layer geometric mapping function is only a function of the observational elevation angle, and large mapping errors will occur in regions with a large electron density or density gradient. In addition, the altitude of the spherical shell in the ionospheric single-layer hypothesis also has a great influence on the results [30]. Therefore, we employ the multi-layer mapping function to reduce the mapping function error with assuming that electrons are distributed on many spherical shells with fixed altitudes from the bottom to the top. Figure 1 shows the schematics of single-layer and multi-layer mapping functions.

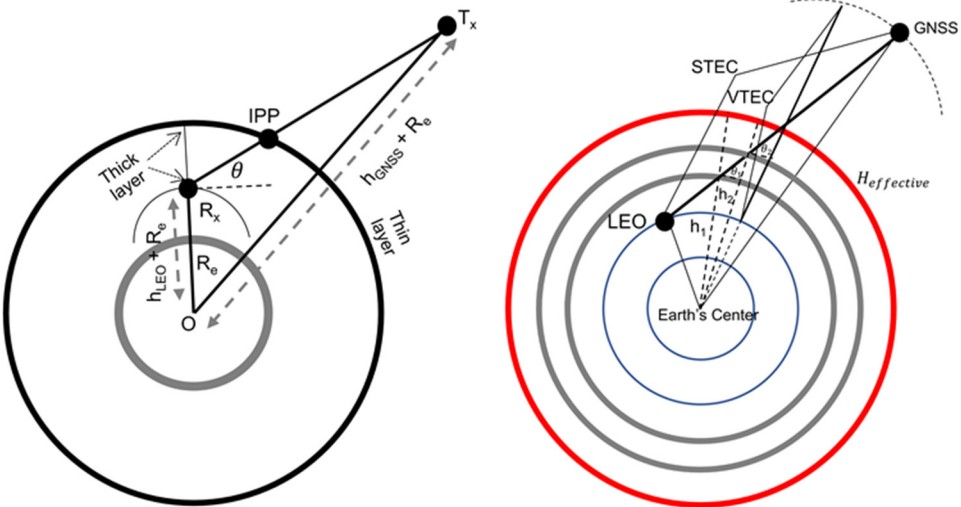

**Figure 1.** Schematics of single-layer and multi-layer mapping functions.

Taking the altitude of 2000 km as the boundary between the ionosphere and the plasmasphere of the Earth, the Chapman function and a superposed exponential decay function are used to describe the electron density distribution of the ionosphere and plasmasphere, respectively [31,32]:

$$n_e^I(h) = N_m exp\left(\frac{1}{2}\left(1 - \frac{h - h_m}{H} - exp\left(-\frac{h - h_m}{H}\right)\right)\right), \tag{1}$$

$$n_e^P(h) = n_p exp\left(-\frac{h}{H_p}\right), \tag{2}$$

where $N_m$ is the peak electron density observed at the peak height $h_m$, $h$ is the altitude from the ground, $H$ is the atmospheric scale height, $n_p$ is the electron density at the bottom of the plasmasphere, $H_p$ (10,000 km) is the mean scale height of the plasma density. These parameters are calculated by Neustrelitz's electron density peak height model (NPHM) and the peak density model (NPDM) [33,34].

Three-dimensional coordinates of LEO and GPS satellites corresponding to each epoch of observation are provided in the podTec file. Assuming that the GPS signal propagates along a straight line, the intersections of the GPS signal and each spherical shell can be calculated using the three-dimensional ray tracing method, which are recorded as $h_1, h_2, \dots, h_n$. The intersection points are projected on a 2D thin-shell surface at an altitude of 450 km and VTEC corresponding to each projection point can be obtained from GIM. According to the vertical distribution structure of electron density, the $vTEC$ between adjacent thin shells can be obtained, denoted as $vTEC_1, vTEC_2, \dots, vTEC_n$. We define the tilted factor corresponding to $vTEC_i$ following [26]:

$$F_i(h_i, \beta) \approx \frac{1}{\sqrt{1 - \left(\frac{(h_i + R_e)cos(\theta_i)}{h_{mIPPi} + R_e}\right)^2}}\left[erf\left(-\frac{exp\left(-\frac{h_i - h_{mIPPi}}{H_{mIPPi}}\right)}{\sqrt{2}}\right)\right]_{h_i}^{h_{i+1}}, \tag{3}$$

where $erf(\ )$ is the error function, $h_i$ and $h_{i+1}$ are the heights of the respective lower and upper shells of $i^{th}$ layer along the ray, $\theta_i$ is the elevation angle of the $i^{th}$ layer, $h_{mIPPi}$ is the peak ionization height, $H_{mIPPi}$ is the atmospheric scale height, and $R_e$ is the average radius of the Earth.

$sTEC_i$ between the $i^{th}$ layer and $(i+1)^{th}$ layer is obtained by multiplying $vTEC_i$ and $F_i(h_i, \beta)$, and $STEC_{model}$ is calculated by adding all the $sTEC_i$ values along the ray. Similarly, the $VTEC_{model}$ is obtained by summation of all $vTEC_i$ values. Thus the conversion factor of the multi-layer mapping function can be defined as:

$$MF_{multilayer} = \frac{STEC_{model}}{VTEC_{model}}, \tag{4}$$

Using the conversion factor $MF_{multilayer}$, the actual slant TEC observations in the podTec file are converted into VTEC. More details about the multi-layer mapping function and the comparison between the multi-layer and single-layer mapping functions can been seen from previous studies [26,35]. Table 1 shows information about the original observational data and the key parameters of the multi-layer mapping function in our data processing.

**Table 1.** Information about the original observational data and the key parameters of the multi-layer mapping function.

| Items | Properties |
|---|---|
| Observational data | Level 1 podTec (.nc) |
| Lower boundary of observation | Orbital altitudes of GRACE and MetOp satellites (~400 km and ~820 km, respectively) |
| Upper boundary of observation | Orbital altitude of GPS satellites (~20,200 km) |
| Cut-off elevation angle | 40° |
| Sampling rate | 10 s |

**Table 1.** *Cont.*

| Items | Properties |
|---|---|
| Boundary between ionosphere and plasmasphere | 2000 km |
| Thin shell interval in ionosphere | 50 km |
| Thin shell interval in plasmasphere | 200 km |

## 3. Results and Analysis

### 3.1. Interplanetary and Geomagnetic Indexes

Geomagnetic storms are closely related to solar activities, and intense geomagnetic storms are more likely to occur during the peak period and downslope of solar activity. From 2007 to 2010, during the solar minimum, there were almost no large geomagnetic storms. Several intense geomagnetic storms took place in 2015 just in the transition from the solar maximum 24 to the downslope. In this paper, we investigate the intense geomagnetic storm on 22 June 2015, and analyzed the responses of VTEC with different altitude ranges during the storm.

Figure 2 shows the variations of the interplanetary parameters and geomagnetic parameters from June 20 to 24, 2015, in which the shaded region represents the time from the beginning of the geomagnetic storm to the end of the main phase. These indices can be obtained from NASA's official website (https://spdf.gsfc.nasa.gov) and the International Service of Geomagnetic Indices (ISGI) website (http://isgi.unistra.fr/data_download.php). Subgraph (a) is the meridional (Bz north–south) component of the IMF, which was observed by the Advanced Composition Explorer (ACE) at the L1 point ~1.5 million kilometers above the Earth. The continuous (more than three hours) southward IMF ($B_Z < 0$) is considered as an important precursor of geomagnetic storms [36]. When the IMF is southward, the momentum and energy carried by the solar wind will be transferred to the Earth's magnetosphere and ionosphere through the mechanism of magnetic reconnection between the IMF and the Earth's magnetic field [37]. Subgraph (b) is the velocity parameter of the solar wind. An obvious staged increase can be seen in the $V_p$ parameter, which was related to interplanetary shocks (IS) of different intensities. Subgraph (c) presents the Auroral Electrojet (AE) indices during the geomagnetic storm. Subgraph (d) shows the symmetric component of the ring current (SYM-H index). With a temporal resolution of 1 min, the SYM-H index can be regarded as a high-resolution Dst index, which is an important parameter reflecting the magnitude of geomagnetic storms. At 18:30 UT (Universal Time) on 22 June 2015, the SYM-H index suddenly increased, marking the Storm Sudden Commencement (SSC). The SYM-H index reached a minimum value of -208 nT at 04:28 UT on June 23, which marked the end of the main phase of the geomagnetic storm. As shown in subgraph (e), the geomagnetic activity index Kp reached a maximum of 8+ during the storm.

### 3.2. Ionospheric Responses From Ground Observations

The main formation mechanism of the ionosphere is that the ultraviolet rays and X-rays in solar radiation enhance the ionization of molecules or atoms in the atmosphere. Therefore, the distribution of ionospheric VTEC is greatly affected by the incidence angle of solar radiation, which is closely related to the local time and latitude. When analyzing responses of the ionospheric VTEC during the geomagnetic storm, the distribution of VTEC on a geomagnetic calm day was taken as a reference value. In our study, the low and middle latitude region between 110°E and 130°E was selected as the research region, and the local time of 120°E was selected as the local time of our research region. The GIM products provided by UPC from June 20 to 24, 2015, were used to extract the VTEC values of the research area. A grid model of VTEC varying with the universal and local time was established with the latitude resolution of 2.5° and the time resolution of 1 h. To more intuitively describe the variation characteristics of ionospheric VTEC during the geomagnetic storm, GIM products of five geomagnetic quiet days (June 15 to 19) before the geomagnetic storm event were used to extract the reference VTEC values by the same method.

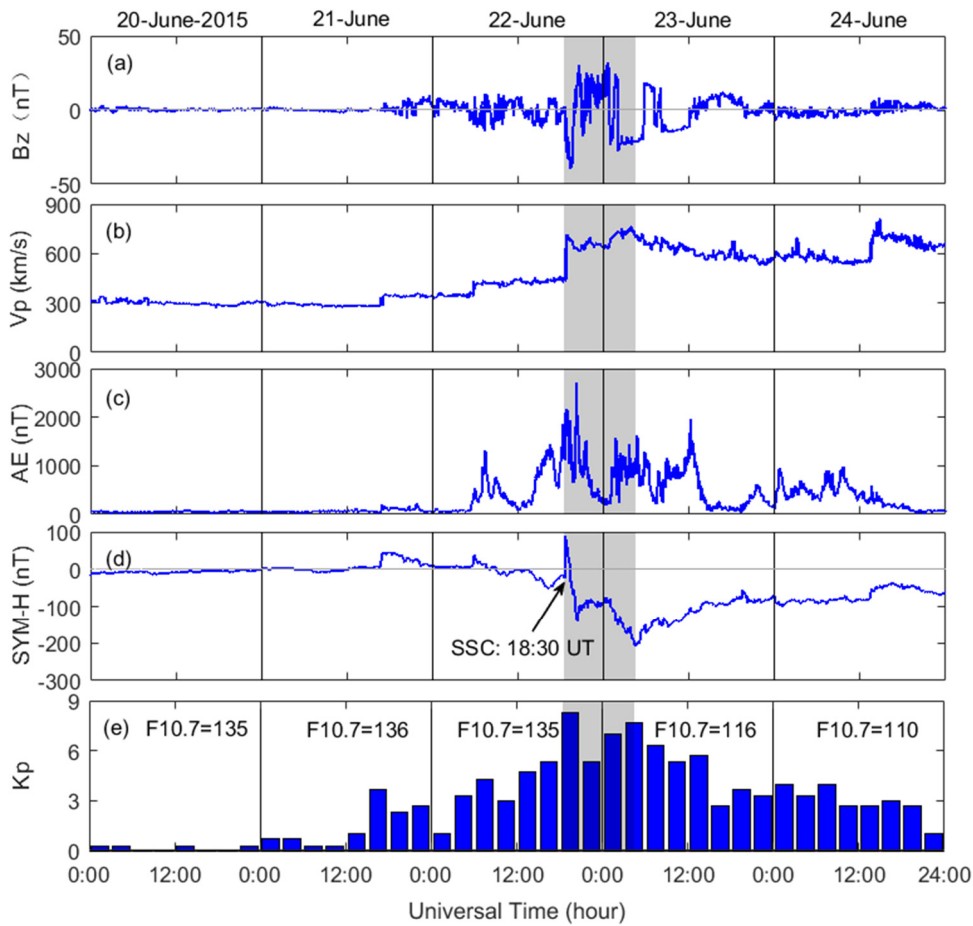

**Figure 2.** Interplanetary and geomagnetic indexes from June 20 to 24, 2015. (**a**) The interplanetary magnetic field $B_Z$ component (nT), (**b**) Solar wind speed $V_p$ (km/s), (**c**) AE index (nT), (**d**) SYM-H index (nT), and (**e**) Kp index.

Figure 3 shows the variations of ionospheric VTEC and deviations of VTEC from the reference values in the low and middle latitude region along ~120°E from June 20 to 24, 2015. The purple dashed lines represent the moments of SSC and the end of the main phase, and the corresponding times are 18:30 UT on June 22, and 04:28 UT on June 23, 2015, respectively. According to the local time, one day is divided into daytime and nighttime. From 08:00 to 20:00 LT (Local Time) is the daytime, while the nighttime is from 20:00 to 08:00 LT of the second day. Since the geomagnetic storm occurred on June 22, which is around the summer solstice in the Northern Hemisphere, and the direct point of solar radiation is near the Earth's Tropic of Cancer (23°26'N), the latitude of the ionospheric peak VTEC is shifted northward as a whole.

In subgraph (a), the most obvious phenomenon is that the maximum value of VTEC appeared within 1–2 h after the end of the main phase. On the one hand, the corresponding local time was about 14:00, when the radiation intensity was at the maximum value on a day, and the number of electrons produced by ionization was the largest. On the other hand, the ionospheric VTEC was significantly higher than the reference value on quiet days due to the super fountain effect caused by the eastward PPEF in the daytime. On event days, the EIA was enhanced, which was related to the eastward PPEF, while in the recovery phase, the formation of EIA was suppressed by the strong westward DDEF in the daytime. Combined with the distribution of dTEC in subgraph (b), a significant positive storm could be seen in the equatorial and low latitude region in the daytime of June 23, while an obvious negative storm was observed in the same region on June 24.

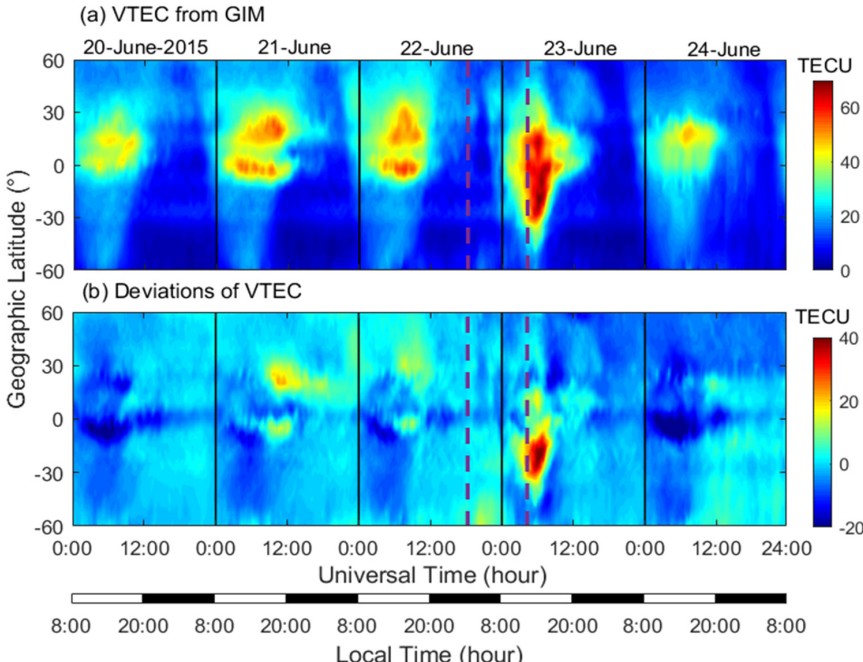

**Figure 3.** Variations of ionospheric VTEC (**a**) and deviations of VTEC (**b**) from the reference values in the low and middle latitude region along ~120°E from June 20 to 24, 2015.

Ground-based GNSS observations can estimate VTEC of the entire ionosphere, namely the vertical integration of the electron density. The profile information of electron density cannot be directly obtained from the GNSS observations, so it is impossible to study responses of the ionosphere at different altitudes. In order to make up for the deficiency of ground GNSS observations, the observational data of Ionosonde were used for supplementary research. For the ~120°E region, four uniformly distributed Ionosonde stations in China were selected, namely Sanya station (109.4°E, 18.3°N), Wuhan station (114.6°E, 30.5°N), Beijing Ming Tombs station (116.2°E, 40.3°N), and Heilongjiang MoHe Station (122.4°E, 53.5°N), which are marked on the map of mainland China in Figure 4. The digital ionosonde can continuously monitor the ionospheric parameter variation over the station. Its observations are of high reliability, but there will also be some obviously erroneous results, which need to be screened.

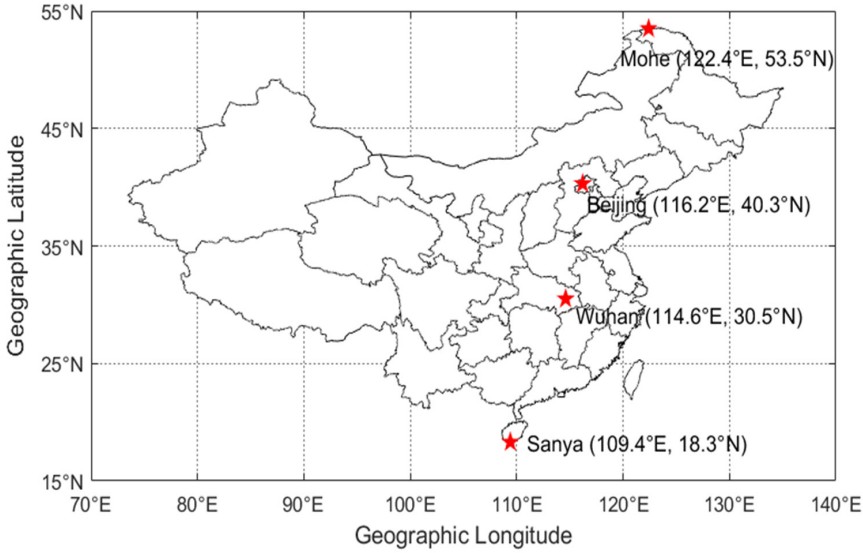

**Figure 4.** Geographical distribution of Ionosonde stations.

Using the observational data of digital ionosondes from June 20 to 24, 2015, we obtained the hmF2 and NmF2 parameters of the ionospheric F2 layer over the Ionosonde stations during the geomagnetic storm. At the same time, the reference values of the hmF2 and NmF2 parameters on the geomagnetic calm day were calculated from observational data of five quiet days before the storm, i.e., June 15 to 19, 2015. Variations in hmF2 and NmF2 are shown in Figures 5 and 6, respectively. The blue dots represent the actual observations during the geomagnetic storm, and the red curves are the reference values of quiet days. Due to the high resolution of the observations, we smoothed the curves of the reference values. The purple dashed lines represent the moments of SSC and the end of the main phase. The black bars at the bottom of the subgraphs indicate the nighttime of each station, i.e., the period from 20:00 to 08:00 LT. Since the four stations are all around 120°E, their local time is very close, with a difference of no more than 1 h. Affected by the intense geomagnetic storm, there was no effective observation on the nighttime of June 22 at the low-latitude Sanya station, which is reflected in subgraphs (d) of Figures 5 and 6, i.e., the missing segments of the blue dots at corresponding periods.

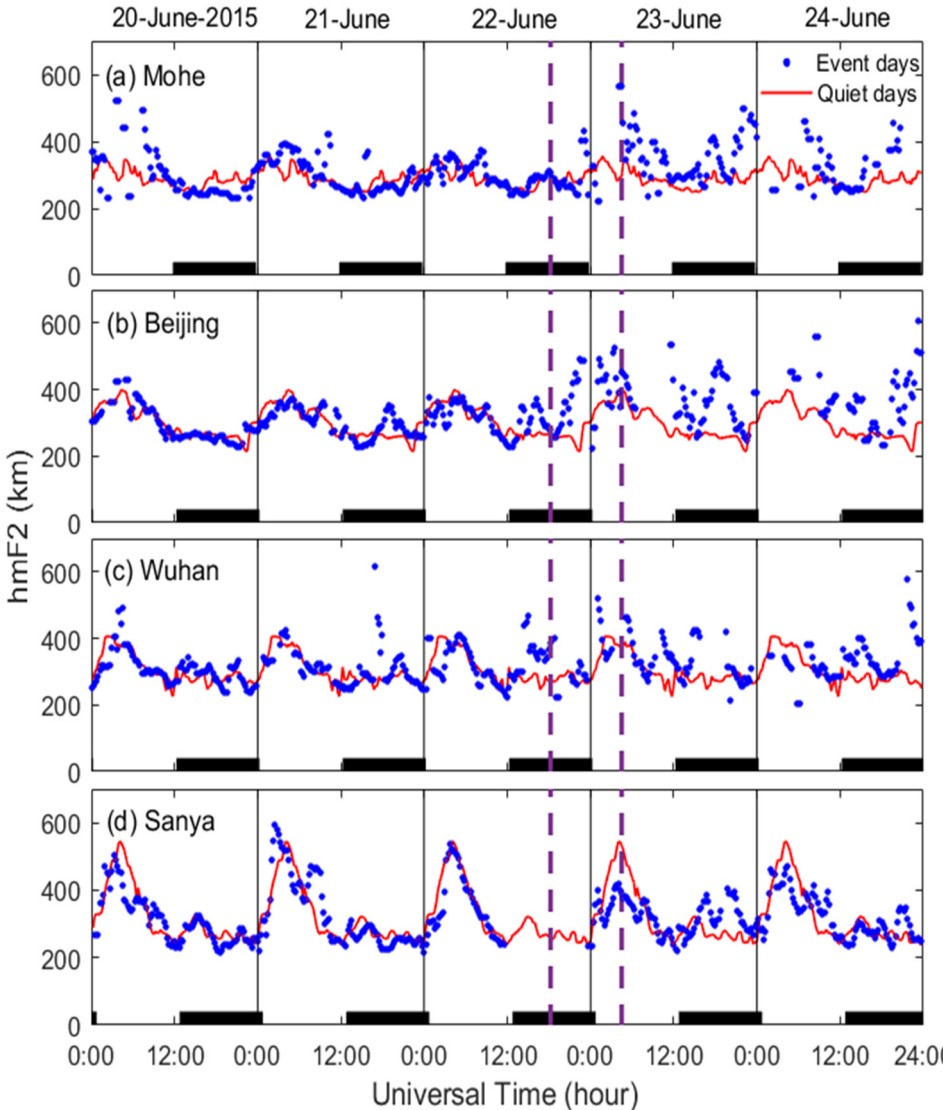

**Figure 5.** Variations of the ionospheric hmF2 over four Ionosonde stations from June 20 to 24, 2015.

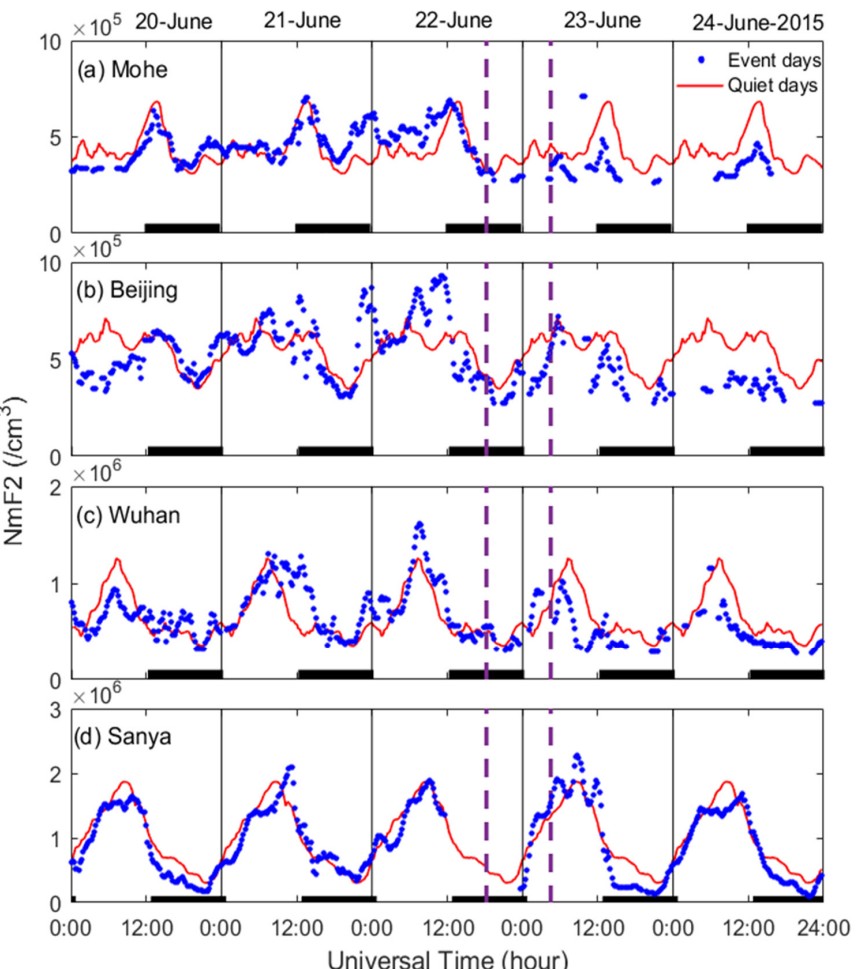

**Figure 6.** Variations of the ionospheric NmF2 over four Ionosonde stations from June 20 to 24, 2015.

From the observational results of the digital ionosondes, it can be seen that there are different ionospheric responses to the intense geomagnetic storm during the daytime and nighttime, and ionospheric responses are also different for observations from different latitudes. For ionospheric hmF2 on geomagnetic calm days, the variation range in the middle latitude region was between 200 km and 400 km, which was smaller at higher latitude. For Sanya station in the low latitude region, the maximum value of hmF2 in the daytime was more than 500 km. Taking the observations of geomagnetic calm days as a reference, in the nighttime of the main phase, significant uplifts were observed at Wuhan and Beijing stations, but a similar phenomenon was not observed at Mohe station with higher latitude, while there was no effective observation at Sanya station at a lower latitude during the period. During the recovery phase, ionospheric hmF2 values over four stations were all uplifted, especially in the nighttime, and observations were relatively sparse, which was due to the poor observational condition caused by the geomagnetic storm. For ionospheric NmF2, a significant increase was observed at the three stations in the middle latitude on June 23, while there was no obvious change at Sanya station at lower latitude. During the recovery phase, the observational results of four stations all showed that the NmF2 was decreased, which was consistent with the effect of the westward DDEF.

### 3.3. Ionospheric Responses From Ground- and Satellite-Based Observations

According to the observational results of digital ionosondes, the intense geomagnetic storm had an impact on the ionospheric parameters hmF2 and NmF2, and therefore VTEC with different altitude ranges should have different responses to the geomagnetic storm. In order to carry out the research, we chose the topside GPS observations of GRACE and MetOp satellites to obtain the VTEC above

their orbits. The orbital altitude of GRACE was about 400 km in June 2015, which was very close to the hmF2 value of the ionosphere, and the up-looking VTEC might be sensitive to the change in hmF2. Topside observations of MetOp satellites with higher orbital altitude (~820 km) were selected for comparative analysis.

The orbits of GRACE and MetOp satellites are both near-polar orbits, and their orbital inclinations are 89° and 98.7°, respectively, which make the local time corresponding to their observations quite concentrated, especially for observations of the middle- and low-latitude region. As shown in Figure 7, for the middle and low latitudes, the local time corresponding to the topside observations of GRACE is concentrated at 10:00–11:00 LT in the daytime and 22:00–23:00 LT in the nighttime, that is to say, the middle and low latitudes observations for different orbits have nearly the same local time, and observations for one orbit are within one hour. Thus, the effect of local time on VTEC does not need to be considered. The deviation between the orbital inclination of the MetOp satellite and 90° is slightly larger, and the local time corresponding to its observations is slightly decentralized, but it is also concentrated at 08:00–11:00 LT in the daytime and 20:00–23:00 LT in the nighttime. The difference in local time corresponding to the topside observations of GRACE and MetOp satellites is within two hours, and both are in the morning or before midnight. The local time has little effect on VTEC. Therefore, the topside observations of GRACE and MetOp satellites can be used to study the differences and similarities in the responses of VTEC with different altitude ranges during the intense geomagnetic storm. Although both the GRACE and MetOp series had two LEO satellites in orbit during the geomagnetic storm period, the DCB of the GRACE-B satellite could not be determined. Its observations could not be used and were eliminated in the data processing. It can be seen from Figure 7 that there were more observations of MetOp than of GRACE, although the color in the figure does not represent the absolute number of observations, which is related to the sampling rate and the resolution of the grid that we set.

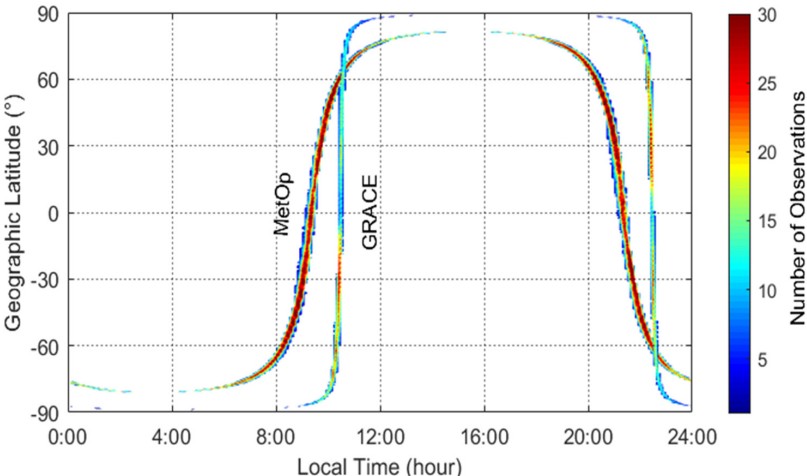

**Figure 7.** Latitude and local time distribution of topside observations of GRACE and MetOp satellites on June 20, 2015.

Figure 8 shows the comparison of the up-looking VTEC from GRACE and MetOp satellites flying over the region along ~120°E with corresponding ground-based VTEC during the daytime and nighttime on the quiet day before the storm onset. Since observations of GRACE are more concentrated, the observations between 80°S and 80°N were selected, while for the MetOp satellite, only the observations between 60°S and 60°N were selected in order to eliminate the influence of local time. In order to accurately determine the orbits of LEO satellites, the GPS receivers on LEO satellites will receive signals from multiple GPS satellites at the same time, and there are two LEO satellites in MetOp series. The up-looking VTEC of one orbit shows the characteristics of multiple observations. The mapping function used to convert the slant path TEC above LEO satellites to VTEC

is the multi-layer mapping function given in Section 2.2, and the cut-off elevation angle is set as 40°. Although the conversion result of the multi-layer mapping function is better than the commonly used single-layer geometric mapping function, it is still unable to unify the converted VTEC from observations of different GPS satellites, especially at the latitude with a peak value of VTEC in the daytime. The topside observations on GPS satellites in northern and southern directions are different, and this is reflected in subgraph (b) that the up-looking VTEC from GRACE shows a deviation of about 5 TECU at the latitude with a peak value of VTEC, which is accordance with the actual situation. There are also a small number of up-looking VTECs with larger values than the corresponding ground-based VTEC, especially for GRACE with lower orbital altitude and the Southern Hemisphere with smaller VTEC values. This is probably due to the problems existing in the observations themselves, which can be ignored since there is currently no good solution.

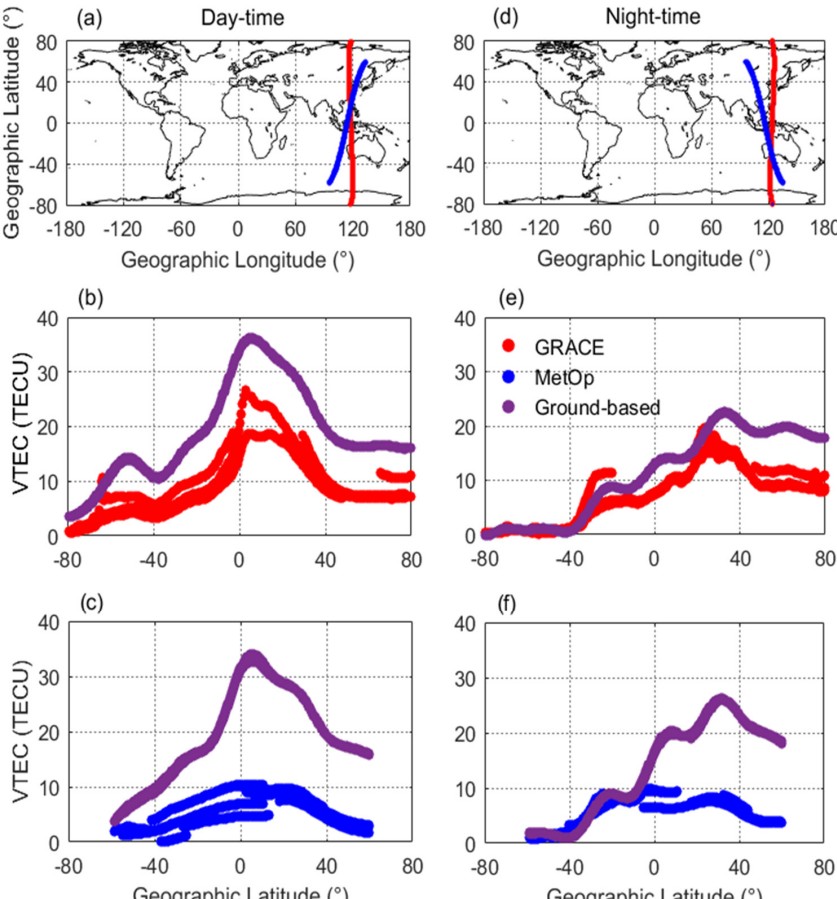

**Figure 8.** Projections of GRACE and MetOp satellites flying over the ~120°E region on the Earth's surface in the daytime and nighttime (**a**,**d**), comparison of the up-looking VTEC from GRACE and MetOp satellites with corresponding ground-based VTEC during the daytime on June 20, 2015 (quiet day) (**b**,**c**) and observations in the nighttime (**e**,**f**).

Due to the lower orbit of GRACE, the distribution of up-looking VTEC from GRACE is closer to that of ground-based VTEC, while the distribution of MetOp with a higher orbit is relatively independent. The up-looking VTEC from GRACE is larger than that from MetOp. The difference in up-looking VTEC between them at the latitude with a peak value of VTEC is 10–15 TECU in the daytime, and the difference is within 10 TECU in the nighttime. In addition, since the geomagnetic storm occurred around the summer solstice in the Northern Hemisphere, the distribution of VTEC in latitude was not symmetrical about the equator, but shifted to the north latitude, which can also be seen in Figure 3. Therefore, in the nighttime Southern Hemisphere, the up-looking VTEC from LEO satellites

was almost equal to the ground-based VTEC, while in the Northern Hemisphere, the ground-based VTEC was significantly higher than the up-looking VTEC from LEO satellites.

Figure 9 shows the comparison of the up-looking VTEC from GRACE and MetOp satellites flying over the ~120°E region with corresponding ground-based VTEC in the daytime and nighttime during the event day on June 22. The contents represented by each subgraph are the same as those in Figure 8, except for the date of observations. Compared with Figure 8, the most obvious phenomenon is that the peak value of the up-looking VTEC from GRACE has increased, almost reaching the level of ground-based VTEC, and the corresponding latitude is about 30°N, which can be related to the uplift of hmF2 observed by the digital ionosonde at Wuhan station. As shown in Figure 5, the hmF2 increased from the reference value of ~300 km to about 400 km in the nighttime on June 22, while the orbital altitude of GRACE was exactly ~400 km. Therefore, the up-looking VTEC from GRACE increased significantly compared with the geomagnetic calm day, which might be related to the eastward PPEF. With respect to the daytime observations in subgraphs (b) and (c), it could be seen that the latitude range corresponding to the peak value of VTEC from both the satellite-based and ground-based observations was wider than that in Figure 8, which showed a certain fountain effect. However, since the daytime orbits of GRACE and MetOp satellites are both "morning orbits", and the EIA phenomenon mainly occurs after noon, the "super fountain effect" caused by the eastward PPEF could not be detected by the topside observations of GRACE and MetOp satellites. LEO satellites with "afternoon orbits" such as COSMIC and Chinese Fengyun series can be considered for further study in the future.

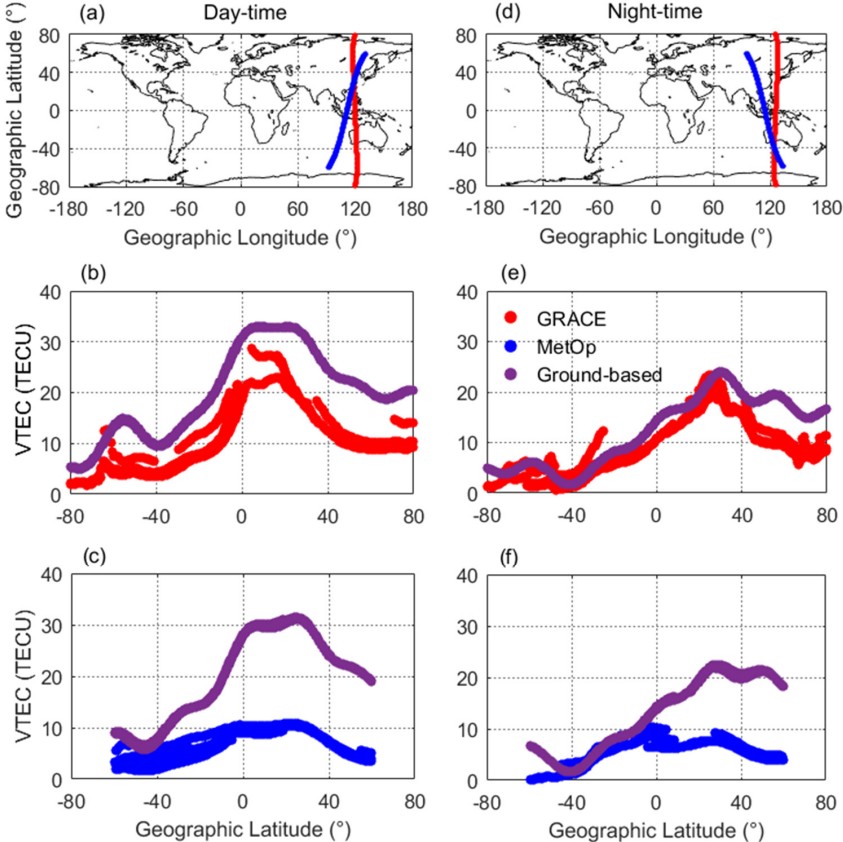

**Figure 9.** Projections of GRACE and MetOp satellites flying over the ~120°E region on the Earth's surface in the daytime and nighttime (**a**,**d**), comparison of the up-looking VTEC from GRACE and MetOp satellites with corresponding ground-based VTEC in the daytime on 22 June 2015 (event day) (**b**,**c**) and observations in the nighttime (**e**,**f**).

Figure 10 shows the comparison of the up-looking VTEC from GRACE and MetOp satellites flying over the ~120°E region with corresponding ground-based VTEC in the daytime and nighttime during the recovery phase on June 24. Compared with Figures 8 and 9, for the daytime observations in subgraphs (b) and (c), the distributions of VTEC were very similar to those on the quiet day before the storm onset, and no significant decreases were observed, indicating that there was no significant negative storm in the daytime. However, for the nighttime observations in subgraphs (e) and (f), the peak values of ground-based VTEC were smaller than those on June 20 and 22, which was related to the fact observed in Figure 6 that NmF2 was lower than the reference value during the recovery phase. The significant negative storm in the nighttime was caused by the strong westward DDEF during the recovery phase. Comparing the up-looking VTEC from GRACE and MetOp satellites during the intense geomagnetic storm, it can be found that the up-looking VTEC from MetOp show little change throughout the storm, while significant variations are observed from the up-looking VTEC of GRACE with lower orbital altitude in the main phase and recovery phase of the storm. Due to the fact that the up-looking VTEC from MetOp is relatively small, its variation is not obvious, while the up-looking VTEC from GRACE with lower orbital altitude is more sensitive to hmF2, and more response characteristics can be detected from the topside observations during the geomagnetic storm.

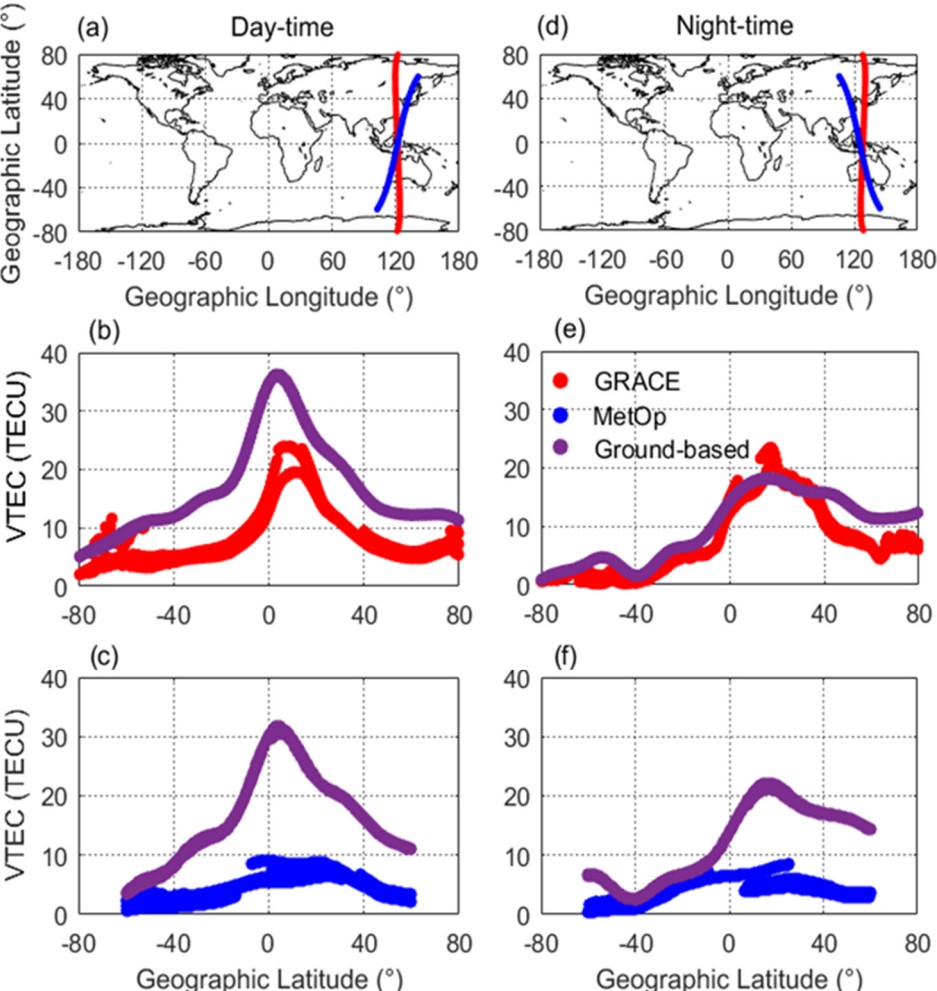

**Figure 10.** Projections of GRACE and MetOp satellites flying over the ~120°E region on the Earth's surface in the daytime and nighttime (**a,d**), comparison of the up-looking VTEC from GRACE and MetOp satellites with corresponding ground-based VTEC in the daytime on June 24, 2015 (recovery phase) (**b,c**), and observations in the nighttime (**e,f**).

Since the topside observations of GRACE and MetOp satellites are for the same longitude region and the local time of the observations from the two LEO satellites is quite close, their observed results can be compared and analyzed. We calculated the differences between ground-based VTEC and the up-looking VTEC from GRACE and MetOp satellites, i.e., the bottom-side VTEC below the orbits of LEO satellites, and the results in the daytime and nighttime during the geomagnetic storm are shown in Figure 11. Although the ground-based VTEC was occasionally smaller than the up-looking VTEC in the Southern Hemisphere observations, only the positive values of dTEC are shown in the figure for practical reasons. Since the orbital altitude of GRACE was lower than that of MetOp satellites, the bottom-side VTEC of GRACE should be smaller than that of MetOp, which was very obvious in the Northern Hemisphere, but the same results were not observed in the Southern Hemisphere with smaller VTEC values. Different from the up-looking VTEC of LEO satellites, the responses of the bottom-side VTEC from MetOp satellites were more obvious during the geomagnetic storm, which also indicated that the ionospheric responses to the geomagnetic storm were concentrated below the orbital altitude of MetOp satellites. The differences between the blue and red lines in Figure 11 can be regarded as the VTEC between the orbital altitudes of GRACE (~400 km) and MetOp (~820 km) satellites. In the daytime observations, it can be seen that the differences between them reached the maximum value during the event day, while for the nighttime observations, the maximum value was reached during the recovery phase on June 24, which was directly related to the response of hmF2 during the storm.

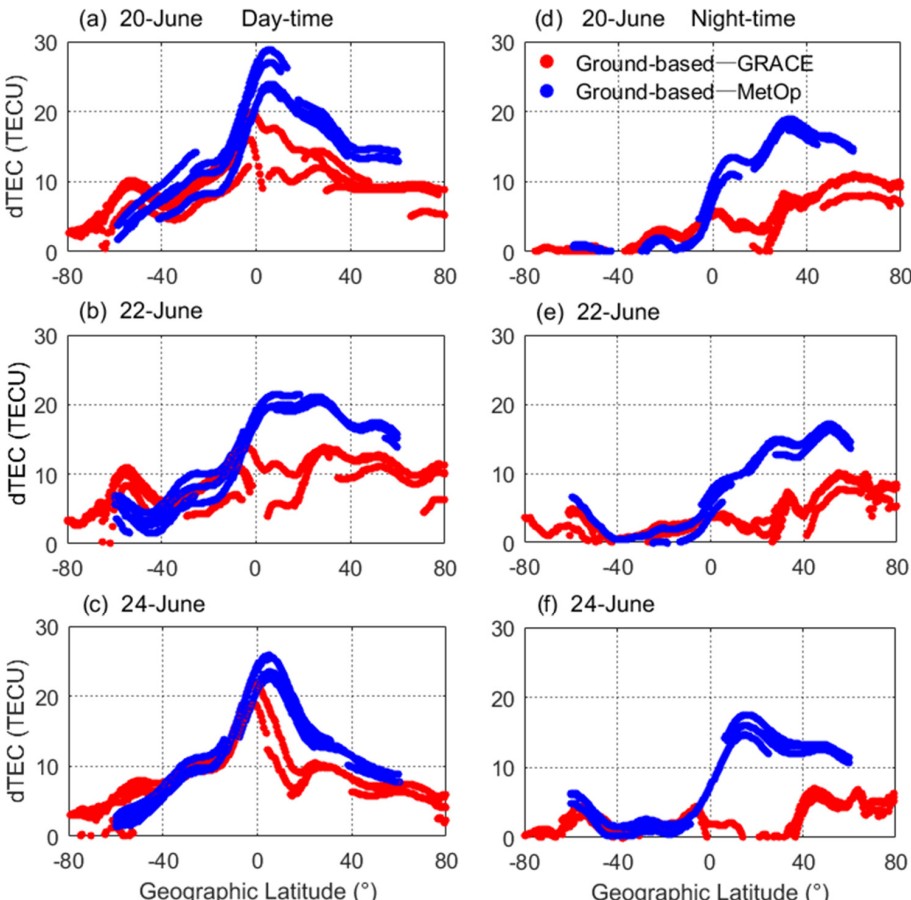

**Figure 11.** Differences between ground-based VTEC and the up-looking VTEC from GRACE and MetOp satellites in the daytime (**a–c**) and nighttime (**d–f**) during the geomagnetic storm.

## 4. Conclusions and Discussion

In this paper, ground- and satellite-based GNSS observations as well as observational data of digital ionosondes are used to study the responses of VTEC with different altitude ranges to the intense geomagnetic storm that occurred on 22 June 2015, which is the second largest geomagnetic storm in the 24th solar cycle. Using the ionospheric product GIMs provided by UPC, the ground-based VTEC in the low and middle latitude region along ~120°E is extracted during the geomagnetic storm, which is compared with the reference value of VTEC on geomagnetic quiet days. Significant positive storms are observed in the low-latitude region during the daytime on June 23. An intense EIA took place at about 06:30 UT along ~120°E on June 23, especially in the Southern Hemisphere. This might be related to the sudden direction change of the PPEF, which was mentioned in some previous studies on the June geomagnetic storm [38–40]. Observational data from four digital ionosonde stations in the research region are used to obtain the hmF2 and NmF2 parameters during the geomagnetic storm, and responses of the ionospheric parameters to the geomagnetic storm at different latitudes during the daytime and nighttime are also analyzed and confirmed.

Furthermore, the responses of the up-looking VTEC from LEO satellites with different orbital altitudes using the topside observations of GRACE and MetOp satellites are also studied during the intense geomagnetic storm. In the process of converting the slant path TEC into the vertical TEC, we employ a multi-layer mapping function, which can effectively reduce the overall deviation caused by the single-layer geometric assumption, where the horizontal gradient of the ionosphere is not considered. Responses of VTEC with different altitude ranges to the storm are studied, and our results show that the topside observations of GRACE with lower orbit (~400 km) could intuitively detect the impact caused by the fluctuation of hmF2, which confirmed the uplift of hmF2 observed by the digital ionosonde during the nighttime on the event day. The latitude range corresponding to the peak value of the up-looking VTEC during the daytime of the event day widened, which is a precursor of the EIA. Unfortunately, the daytime orbits of the GRACE and MetOp satellites are both "morning orbits", and therefore the super fountain effect caused by the eastward PPEF could not be detected. On the other hand, for the topside observations of MetOp satellites with higher orbital altitude (~820 km), no obvious response of the up-looking VTEC was observed during the geomagnetic storm. However, for more intense geomagnetic storms or local afternoons when the variation of VTEC is more obvious, ionospheric responses to the geomagnetic storm might be detected from the topside observations of LEO satellites with higher orbits.

**Author Contributions:** C.G. processed the data, performed the analysis, and wrote the article. S.J. provided supervision, performed the analysis, and reviewed the article. L.Y. Implemented and evaluated the multi-layer mapping function. All co-authors helped discussing and reviewing the article. All authors have read and agreed to the published version of the manuscript.

**Funding:** This study is supported by the National Natural Science Foundation of China (NSFC) Project (Grant No. 41761134092) and Shanghai Leading Talent Project (Grant No. E056061).

**Acknowledgments:** We thank the following organizations for providing the data used in this work: CSERF, UPC, CDAAC.

**Conflicts of Interest:** The authors declare no conflict of interest.

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
