# Peer review of "Ionospheric Responses to the June 2015 Geomagnetic Storm from Ground and LEO GNSS Observations"

_remotesensing, doi:10.3390/rs12142200_

Round 1

Reviewer 1 Report

In this paper, the authors studied the impact of geomagnetic storm on the ionospheric effects. The result seems correct. It would be great if the authors may provide some results on how the ionospheric storm can affect the positioning accuracy of GNSS systems.

Author Response

  Thanks very much for your comments and suggestion. This paper was mainly focused on ionospheric responses to the June 2015 geomagnetic storm from ground and LEO GNSS observations. In the future we will further investigate its effect on GNSS positioning.

Reviewer 2 Report

This paper uses a good conjunction of sensors at various altitudes to examine the ionospheric response to a substantial geomagnetic storm, so the work is somewhat unique in this regard. The work is presented in a good coherent and confident style of writing so the argument is clear for the reader. The experiment design and methodology is well considered in the context of existing literature and the Results and Analysis is very thorough.

However, the critical analysis of the results compared with the existing literature is lacking in the Conclusions, and this needs to be addressed. The Last section could be re-named Discussion and Conclusions and the addition of 2 or 3 paragraphs comparing and contrasting results with some of the papers in References (e.g. 3-5, 7, 8, 12-20) should be able to achieve this.

Comments and Suggestions;
Page 1 Introduction

Line 1: "The Geomagnetic storm is a severe global disturbance of the Earth's magnetic field, usually under the impact of disturbances in the Solar Wind and Interplanetary Magnetic Field, with their origins near the solar surface.

Remove "A large amount of charged particles carried by CME reach the earth's space and are captured by the geomagnetic field," as this is not strictly correct. CMEs usually have a shock front that is much wider than the cross-section of the magnetosphere so most the particles miss Earth.

L5: "Geomagnetic Storms result in an increase of the ring current, and the magnetic field generated by the ring current is superimposed on the geomagnetic field, which results in great changes in the horizontal component H."

Page 2
Para 1, L2 "...satellites operating in the magnetosphere and ionosphere (in Geostationary, Medium and Low Earth orbits), such as radiation hazard to electronics and affecting communications through the ionosphere."

Para 2, L2: "Ionospheric storms can be divided into positive and negative, ..."

Para 2, L5: "circulation proposed by Duncan [5] and continuously.."

Para 3, L9: "In addition, the PPEF occurring at dusk can..."

Page 7  Results & Analysis
Para 1: "Geomagnetic storms are closely related to solar activity, and intense geomagnetic storms are more likely to occur during the peak period and downslope of solar activity. From 2007 to 2010, during solar minimum, there were almost no large geomagnetic storms. Several intense geomagnetic storms took place in 2015, as that year was just in the tranition from solar maxium 24 to the downslope. In this paper, we selected the intense geomagnetic storm occurring on June 21 as the study case...."
[cycle 24 peak was ~2012-14]

Page 8
Line 3: "Subgraph (a) is the meridional (Bz north-south) component..."
Line 4: "....(ACE) at the L1 point ~1.5 million kilometers upstream of Earth."
Line 5: "...as an important precursor of geomagnetic storms [36]."
Line 10: "... transferred to the earth’s magnetosphere and ionosphere through the mechanism of magnetic reconnection between the IMF and the Earth's magnetic field [reference required]."
[Suggest using an existing reference in the list to avoid re-numbering. Use one that invokes reconnection/merging and cite as [XX and references within]. Alternately a good reference is Cowley, S. W. H., and M. Lockwood (1992), Excitation and decay of solar wind-driven flows in the magnetosphere-ionosphere system, Ann. Geophys., 10, 103 – 115.]
Line 11: "...during the main phase, which is probably related to CME impact on the geomagnetic field."

P10: The paragraph under Fig 5 is in a smaller font ?  Looks like the figure caption font has carried over into the main body text.

P18, Line 1 'Although the ground-based VTEC was occasionally smaller than the up-looking VTEC in the southern hemisphere, only the positive values of dTEC were shown in the figure.'
Ground-based VTEC being smaller than the VTEC measure from a spacecraft should not be physically possible. So this suggests errors (1-4 TECU?) in both measurements are large enough in overlap (e.g. ground underestimate and space overestimate) with a weak TEC to produce a negative dTEC. It would be better to not phrase it as the 'actual dTEC' being negative, but as excluding the cases where the 'measured VTEC with errors' gave non-physical negative dTEC.

P18 Conclusions
As mentioned in the introductory comments, this section needs to be expanded with a critial analysis comparing the results with the literature.

Para 2, line 5: "...geometric assumption where the horizontal gradient of ionosphere..."
Line 13-14 '...MetOp satellites with higher orbital altitude (~820 km), no obvious response was observed during the geomagnetic storm.'
This is an important result that suggests there was no change to the plasmasphere above the topside ionosphere during the storm. This should be discussed and compared with the literature.
The authors should read 'On some features characterizing the plasmasphere–magnetosphere–ionosphere system during the geomagnetic storm of 27 May 2017' by Pezzopane et al, Earth, Planets and Space volume 71, Article number: 77 (2019).

References
All papers cited in the text appear in the References, and all papers in the References are cited in the text.
Good to see some classic literature cited (D.F. Martyn 1953).

Author Response

  Thanks very much for your detailed corrections and constructive comments. We have revised all according to your suggestions in the revised manuscript.

Reviewer 3 Report

Review for “Ionospheric responses to the June 2015 geomagnetic storm from ground and LEO GNSS observations” by Chao Gao et al. The article is devoted to study the ionospheric response on the intense geomagnetic storm of June 22-23, 2015. The authors consider the effects along the meridional chain of stations, using ionosonde measurements and satellite observations. The most interesting part of the work, in my opinion, is related to the study of the ionosphere in various altitude layers. Understanding the reactions of the topside ionosphere (above the peak height) is a challenging task, due to the lack of direct measurements. Therefore, the results of the work deserve attention. The multilayer mapping function, proposed by the authors, is also of interest for many other studies. Therefore, I believe that the article can be published after minor corrections, after the authors answer to some questions that I have. They are listed below.

1 The first and main remark. The June 22-23, 2015 storm is a very famous event, the study of which is performed in a large number of works, including the study of PPEF, DDEF effects on the ionosphere, and the studies of the topside ionosphere. Authors should cite these works and give a comparison with the results obtained earlier. They should emphasize more clearly the novelty of the results obtained. List of key references is attached below:

Astafyeva, E.; et al. Prompt penetration electric fields and the extreme topside ionospheric response to the 22–23 June 2015 geomagnetic storm as seen by the Swarm constellation. Earth Planets Space 2016, 68, 1309, doi:10.1186/s40623-016-0526-x.

Astafyeva, E.; et al. Global Ionospheric and Thermospheric Effects of the June 2015 Geomagnetic Disturbances: Multi-Instrumental Observations and Modeling. J. Geophys. Res. Space Phys. 2017, 122, 11716–11742, doi:10.1002/2017JA024174.

Astafyeva, E.; et al. Study of the Equatorial and Low-Latitude Electrodynamic and Ionospheric Disturbances During the 22–23 June 2015 Geomagnetic Storm Using Ground-Based and Spaceborne Techniques. J. Geophys. Res. Space Phys. 2018, 123, 2424–2440, doi:10.1002/2017ja024981.

Piersanti, M; et al. Comprehensive Analysis of the Geoeffective Solar Event of 21 June 2015: Effects on the Magnetosphere, Plasmasphere, and Ionosphere Systems. Sol. Phys. 2017, 292, 5–169, doi:10.1007/s11207-017-1186-0.

Reiff, P.; et al. Multispacecraft observations and modeling of the 22/23 June 2015 geomagnetic storm. Geophys. Res. Lett. 2016, 43, 7311–7318, doi:10.1002/2016gl069154.

Singh, R.; Sripathi, S. Ionospheric Response to 22–23 June 2015 Storm as Investigated Using Ground-Based Ionosondes and GPS Receivers Over India. J. Geophys. Res. Space Phys. 2017, 122, 645, doi:10.1002/2017ja024460.

Yasyukevich Yu.V.; et al., Small-Scale Ionospheric Irregularities of Auroral Origin at Mid-Latitudes during the 22 June 2015 Magnetic Storm and Their Effect on GPS Positioning Remote Sens. 2020, 12, 1579; doi:10.3390/rs12101579

  1. The considered storm is usually associated with the Halo CME (~1350 km/s) appeared at 2:36 UT on June 22, 2015. Sudden storm commencements (SSC) were registered at ~05:45 and ~18:30 UT on June 22, when the shocks hit the Earth’s magnetosphere (see references listed above). Therefore, it is unclear why the authors take the storm beginning on June 21. Considering the behavior of geomagnetic indices, this day is a quiet period, which might be described as pre-storm conditions.
  2. The authors use in the study GIMs from CODE laboratory with a resolution of 1 hour. Why not to use GIMs from UPG laboratory that have a higher time resolution (15 min), which corresponds to the resolution of ionosonde measurements?
  3. Page 10, the first paragraph: “… obvious negative storms could be seen in the equatorial and low latitude region in the daytime from June 20 to 22…”

How can you see the effect of the storm on June 20 and 21, if the storm started later? This negative effect cannot be related to the storm and needs a different interpretation.

  1. I do not like too bold lines in Figures 9-12. This is especially disturbing in Figure 12, where the red line is completely hidden behind the blue ones in some places.
  2. The conclusion section is mostly a repetition of what has been done. There is not enough description and discussion of the obtained results with emphasis of their novelty.
  3. I am not sure about the correctness of using the IRI model as a standard for the upper ionosphere (Figure 2). IRI is known to work very poorly in the plasmasphere and the upper ionosphere. Would it be better to use IRI-plus for verification? It is not a critical remark, just an offer.
  4. Language:

Page 3, the last paragraph “The satellite-based observations used in this paper are the topside observations of GRACE and MetOp satellites …”

Page 4, paragraph 2 “And the tilted TEC is the absolute value of TEC in which the DCBs of GPS satellite and receiver are removed…”

Author Response

  The details about the response are given in the file uploaded.
